# Integrating XAJ Model with GIUH Based on Nash Model for Rainfall-Runoff Modelling

**Yingbing Chen** [1,2], **Peng Shi** [1,2], **Simin Qu** [1,2,*] , **Xiaomin Ji** [3], **Lanlan Zhao** [4], **Jianfeng Gou** [1,2] **and Shiyu Mou** [1,2]

1 State Key Laboratory of Hydrology-Water Resources and Hydraulic Engineering, Hohai University, No. 1 Xikang Road, Gulou District, Nanjing 210098, China; yingbing.chen1@gmail (Y.C.); ship@hhu.edu.cn (P.S.); jf_gou@sina.com (J.G.); ariesmsy@hhu.edu.cn (S.M.)

2 College of Hydrology and Water Resources, Hohai University, No.1 Xikang Road, Gulou District, Nanjing 210098, China

3 Jiangsu Province Hydrology and Water Resources Investigation Bureau, No. 5 Shanghai Road, Gulou District, Nanjing 220098, China; jixiaominqq@sina.com

4 Bureau of Hydrology, Ministry of Water Resources of the People's Republic of China, Beijing 10053, China; zhaolanlan@mwr.gov.cn

* Correspondence: wanily@hhu.edu.cn

**Abstract:** The geomorphologic instantaneous unit hydrograph (GIUH) is an applicable approach that simulates the runoff for the ungauged basins. The nash model is an efficient tool to derive the unit hydrograph (UH), which only requires two items, including the indices *n* and *k*. Theoretically, the GIUH method describes the process of a droplet flowing from which it falls on to the basin outlet, only covering the flow concentration process. The traditional technique for flood estimation using GIUH method always uses the effective rainfall, which is empirically obtained and scant of accuracy, and then calculates the convolution of the effective rainfall and GIUH. To improve the predictive capability of the GIUH model, the Xin'anjiang (XAJ) model, which is a conceptual model with clear physical meaning, is applied to simulate the runoff yielding and the slope flow concentration, integrating with the GIUH derived based on Nash model to compute the river network flow convergence, forming a modified GIUH model for flood simulation. The average flow velocity is the key to obtain the indices *k*, and two methods to calculate the flow velocity were compared in this study. 10 flood events in three catchments in Fujian, China are selected to calibrate the model, and six for validation. Four criteria, including the time-to-peak error, the relative peak flow error, the relative runoff depth error, and the Nash–Sutcliff efficiency coefficient are computed for the model performance evaluation. The observed runoff value and simulated series in validation stage is also presented in the scatter plots to analyze the fitting degree. The analysis results show the modified model with a convenient calculation and a high fitting and illustrates that the model is reliable for the flood estimation and has potential for practical flood forecasting.

**Keywords:** flood forecasting; GIUH; Nash model; Xin'anjiang model; flow velocity

## 1. Introduction

Flood is one of the most devastating natural disasters on earth, which causes immense human safety damage and property loss worldwide [1], especially in some areas with insufficient hydrologic data, the data scarcity has led to the inability of many hydrologic model applications, causing inaccurate flood prediction and failing the prevention of flood disaster. As an approach of calculating flow concentration, the geomorphologic instantaneous unit hydrograph (GIUH) method is a burgeoning

and valid method, which can be obtained without hydrologic record. Rodriguez-Iturbe et al. [2], Valdes et al. [3], and Gupta et al. [4] initially proposed GIUH to describe the movement of each droplet by using the probability density function. It is based on the Horton–Strahler stream ordering scheme [5–7], with a view at the stochastic distribution of water droplet concentration time. Rodriguez-Iturbe and Valdes [2] applied the theory of probability method to hydrological flow concentration and derived the R-V GIUH and the formula of three-order river network, including the reasonably transferring rules of each droplet. According to the theory, Gupta et al. 1983) [8] improved and modified the expression of GIUH with the statistical method, proposing the probability distribution function of lag time, which is in an exponential function form, and it has provided a feasible way to deal with the distribution of travel time for any drop of rainfall landing in a watershed. Over the same period, Rosso [9] explored the relation between Horton order ratios and the two parameters of the Nash model from in a numerical perspective, specifically explaining the effect of catchment geomorphology on hydrologic response.

After the establishment of a comprehensive formula derivation of GIUH, many hydrologists started to seek the specific improvements that aimed at the GIUH derivation steps. In order to express the influence of the hydrodynamic diffusion on GIUH, the conception of width function is employed to abstract the GIUH [10–12]. Lu [13] adopted the deterministic modeling in accordance with the linear reservoir and the geomorphology of a watershed, into ascertaining GIUH, furthermore proving the idea of the similarity between the Nash unit hydrograph and the GIUH. Choi et al. [14] studied a methodology to estimate the parameters $n$, $k$, on the basis of the geomorphologic dispersion that results from the uneven spatial distribution along the flow path of a watershed. The flow velocity $v$ is the key to obtain the indices $k$.

At the same time, a lot of hydrologists are devoted to investigating the other aspects on the calculation of GIUH. Based on the kinematic wave theory, a new expression of GIUH is derived by studying the travel time of overland and stream channel in the sub-basins of the Keelung watershed [15]. Rui and Shi [16,17] considered the geomorphologic and hydrodynamic dispersion and formed the formula to obtain the watershed geomorphologic instantaneous unit hydrograph (WGIUH) and took the fractal self-similar characteristics into account to derive the corresponding width function of the watershed network to calculate the GIUH. Sarkar and Rai (2011) applied the Soil Conservation Services-Curve Number (SCS-CN) method [18] for rainfall excess estimation and the Nakagami-m distribution was used to compute the GIUH of different sub-catchments of upper Ganga river system. Roger Moussa [19] proposed the definition of several new equivalent indices of Horton–Strahler ratios that were unaffected by the area threshold, which demonstrates a new method to calculate the GIUH with higher certainty. Simultaneously, the GIUH method is revised and widely testified in the runoff simulation. Lee and Chang (2005) optimized the geomorphology-based IUH model, which can reflect the hydraulic conductivity and the roughness of the surface flow [20] to estimate both the surface runoff and the sub-surface flow. Based on such modified model, in 2012, Sabzevari et al. [21] revised to propose a saturation model to predict the surface and the subsurface flow in using the GIUH method in the Kasilian watershed. Kumar. R et al. applied the GIUH model based on Nash and Clark model into the runoff simulation of the ungauged basins using the Horton–Strahler ratios and worked out the uncertainty analysis of the results in 2004 [22], and in 2007 [23], they followed on this topic to investigate the GIUH applied in ungauged areas on runoff flow estimation. Sahoo B. et al. studied the difference between the GIUH that is based on the Clark and Nash model, providing a comparison of the two models on the direct runoff simulation [24]. For further investigation on the GIUH model validation, Irshada Iskender and N. Sajikumar have estimated the surface runoff while using the SWAT model and GIUH, respectively, and compared the model performance, which shows that the GIUH model performance is marginally better than the SWAT model on a daily scale [25]. Kumar. A and Kumar. D [26] applied the GIUH model for the forecasting of the direct runoff hydrograph (DRH) in the hilly watershed, which showed relatively good accuracy. In 2009, Rai et al. [27] explored whether the GIUH can be used as a transfer function to obtain the unit hydrograph (UH) for transferring the

excess rainfall into surface runoff, in which excess rainfall is a production function to the hydrologic system. Kumar. A [28] simulated the hydrologic response of two four-order hilly catchments in the central Himalayan region of India by the GIUH, which is derived from using two different models; one is based on the traditional Horton's stream ratios and the other is based on the two scale parameters of the Nash model.

As demonstrated in those literatures, GIUH is commonly and successfully used in simulating the flood events. However, the GIUH describes the process of a droplet flowing track from the falling position on the land to the basin outlet covering the part of river flow concentration that itself cannot be directly used for comprehensive runoff process simulation. While these aforementioned studies always employ the effective rainfall, generally calculating in an empirical way and scant of accuracy, to convolve with the GIUH, and then evaluate the total flow discharge. In this paper, the GIUH model is modified by coupling it with the Xin'anjiang (XAJ) model to simulate the flood, including both runoff generation and flow concentration in a comprehensive conceptual model with a physical meaning. This modification improves the predictive capability of the GIUH model and, at the same time, it maintains the advantage that the GIUH needs less observed data, increasing the applicability of the model in the ungauged or scantly gauged basins. Combined with the GIUH to simulate the flood events in three hilly basins, the modified model to simulate the flood in data scarcity areas is constructed. The Shaowu, Jianyang, and Shuiji watersheds, which are located in the southeastern China, are selected to examine the feasibility of the model. The performance of developed GIUH model along with the XAJ model are statistically analyzed by 10 flood events for calibration and six for validation.

## 2. Study Area and Data Collection

### 2.1. Study Area

The Minjiang River is located in southeast China, with the longest one in Fujian province, which originates from the east of Wuyi Mountain on the border of Jiangxi and Fujian province and flows as a length of 577 km through more than 30 counties. This watershed lies between 116°23′ E and 119°43′ E longitude and 25°23′ N to 28°19′ N latitude with three main tributaries: Shaxi River, Futunxi River, and Jianxi River. The whole basin is approximately a fan shape that covers an area of 60,992 km$^2$ with abundant natural resources holding the average annual runoff volume at 1980 m$^3$/s ranking at the seventh among the major rivers all around China, containing the percentage of land-covered forests as 59%, basically being classified as pinus massoniana forest, the evergreen broad-leaved forest, Chinese fir, and bamboo forest.

The Minjiang watershed is situated in the subtropical monsoon climate zone with adequate rainfall and sunshine, and the annual average temperature is from 15 °C to 20 °C, while it can be higher than 40 °C in summer. Due to the thermal difference between land and sea, the monsoon climate is shown significantly on the uneven temporal and spatial distribution of precipitation (P) within the regions, keeping the total amount and distribution in accordance with the rainfall condition. The average annual rainfall in the upstream of Minjiang basin varies from 1800 mm to 2600 mm, 1600 mm~1800 mm in the midstream, and 1200 mm~1600 mm in the downstream, and the mean annual evaporation (E) stably fluctuates in contrast, which ranges from 735.5 mm to 1010.5 mm at different locations over the whole area. Basically, the flood period in the study areas occurs from April to September, the plum rainstorms from April to June, and the floods by typhoon principally in the latter three months. Owing to the temporal and spatial heterogeneity of precipitation, the distribution of runoff presents unevenly. Spatially, in the midstream and downstream, the mean annual runoff depth ranges from 700 mm to 900 mm, with a higher value of 1962 mm in the upstream region. Temporally, the runoff in rainy season, accounts for 70% to 80% of the total annual runoff.

Three sub-basins with observed hydrological data are collected in this study, of which the control stations are named Shaowu, Jianyang, and Shuiji, respectively. The study areas are generally located in the upstream region of Minjiang watershed; in other words, the hydro-meteorological characteristics

and the underlying surface conditions meet the description of the upstream region of Minjiang watershed. The Shaowu catchment lies in the far west part of the whole region, with a drainage area of 2677 km$^2$, at the land slope ranging from to 0.0015 to 1.04. Generally speaking, the higher the order of a river, the gentler the reflected channel slope. The total area of the Jianyang catchment is 3253 km$^2$ and the land slope varies from 0.0012 to 0.87. As for the Shuiji watershed, it is the largest one covering an area of 3470.5 km$^2$, with the land slope from 0.0009 to 0.77. Figure 1 shows the delineation maps of the study areas.

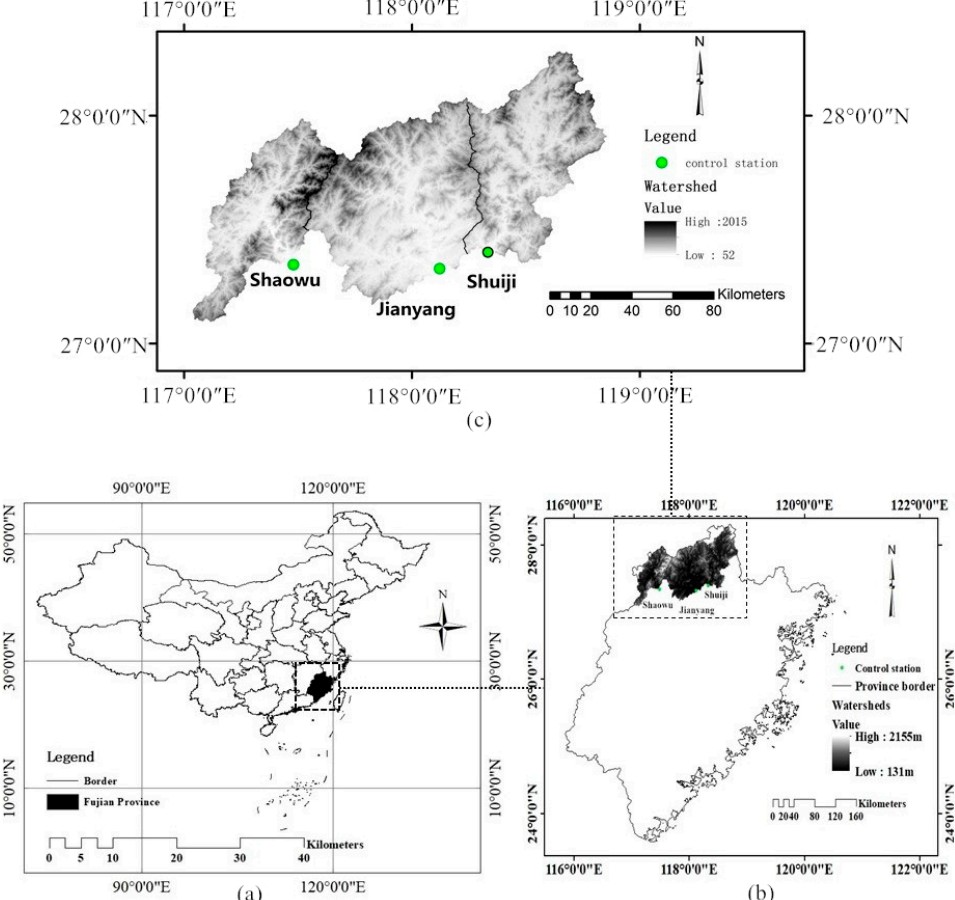

**Figure 1.** Location map of three watersheds selected in the study (the watershed delineation map indicates that, the selected basins (Shaowu, Jianyang, Shuiji) located in the north of Fujian Province in the southeastern of China, and elevation and terrain information of each). (**a**) location of Fujian province in China, (**b**) location of three studied watersheds in Fujian Province, (**c**) watershed digital elevation model (DEM) map of Shaowu, Jianyang, and Shuiji watersheds.

### 2.2. Hydrologic Data and Preprocessing

The observed precipitation, the evaporation, and flow discharge data from 1988 to 1999 are acquired from the hydrological year book of the Minjiang River Basin. Rainfall data are collected from the rain gauge stations in each catchment. The flow discharge during the period is measured from the hydrologic control station of each basin, which is marked in Figure 1. The type of E-601 pan, which is a common tool of the daily evaporation measurement, is still used in the routine measurement.

The division of river channel order is based on the Horton–Strahler stream ordering scheme, and the characteristic values of the geomorphology and terrain, like the channel length ratio, bifurcation ratio, and the area ratio of each region are all based on the stream channel ordering, extracted based on the Aster GDEM data at the resolution of 30m (downloaded on "gscloud. cn"). Table 1 provides all of the geomorphologic and topographic parameters in need.

**Table 1.** The Horton geomorphologic indices and basic topographic values in Shaowu, Jianyang, and Shuiji basin.

| Watershed | $S_0$ (km$^2$) | $L_\Omega$ (km) | Stream Area Ratio, $R_A$ | Bifurcation Ratio, $R_B$ | Stream Length Ratio, $R_L$ |
|---|---|---|---|---|---|
| Shaowu | 2677 | 49.74 | 4.2 | 3.99 | 2.13 |
| Jianyang | 3253 | 72.77 | 4.292 | 4.345 | 2.211 |
| Shuiji | 3470.5 | 86.93 | 4.326 | 4.209 | 2.187 |

## 3. Methodology

### 3.1. XAJ Model

The Xin'anjiang (XAJ) model was initially proposed for the prediction of Xin'anjiang Reservoir inflow and flood forecasting by Zhao Renjun [29,30]. The complete calculation process of the XAJ model is made up of four parts: the evaporation module, the runoff yielding module, the runoff sources partition module, and the runoff concentration module.

In the XAJ model, the non-uniform vertical distribution of soil is taken in account, whereby the three-layer calculation algorithm is applied to compute total evapotranspiration:

(1)　when $WU + P \geq EP$,

$$EU = EP, \ EL = 0, \ ED = 0 \tag{1}$$

(2)　when $WU + P < EP$ and $WL \geq C \times WLM$,

$$EU = WU + P, \ EL = (EP - EU) \times \frac{WL}{WLM}, \ ED = 0 \tag{2}$$

(3)　when $WU + P < EP, C \times (EP - EU) \leq WL < C \times WLM$,

$$EU = WU + P, \ EL = C \times (EP - EU), ED = 0 \tag{3}$$

(4)　when $WU + P < EP, \ WL < C \times (EP - EU)$,

$$EU = WU + P, EL = WL, ED = C \times (EP - EU) - EP \tag{4}$$

where, EP is the evapotranspiration capacity (mm); WU, WL are respectively the upper and lower soil moisture (mm); P is the precipitation(mm); C is the deep layer dispersion coefficient; WLM is the soil moisture storage capacity of lower layer (mm); EU, EL, ED are the soil evaporation of upper, lower and deep layer (mm).

As for this study, the XAJ model was applied to compute the total inflow to river network, combined with the GIUH method that is based upon the Nash model to calculate the runoff process. The specific parameters corresponding to each part of calculation are listed in Table 2, including the physical meaning, the suggested general range of every parameter, and the calibrated value in each catchment. Figure 2 depicts the complete operation process of the XAJ model. Referring to the concept of hillslope hydrology, the XAJ model was modified to a 3-water-source partition principle to calculate the runoff generation. A conceptual structure of free water reservoir is considered according to the vertical distribution of soil moisture. Through the free water reservoir, the total runoff is separated into the three different components: surface runoff, interflow, and groundwater runoff, stated in Figure 3. "R" represents the total runoff based on the theory of the runoff yield in saturated zone, and there are 2 outlets of the reservoir on the runoff yield region, one of which is a lateral exit forming the subsurface runoff "RS"; the other one is the gate of ground water "RG".

**Table 2.** Parameter of the XAJ model (in the sensitive degree column, S stands for sensitive and I for insensitive; the range of every parameter is for reference, and the corresponding values of Shaowu, Jianyang, Shuiji catchments are presented in the list are all under calibration).

| Module | Parameter | Physical Significant (Unit) | Sensitive Degree | Range | Shaowu | Jianyang | Shuiji |
|---|---|---|---|---|---|---|---|
| Evaporation | KC | potential evaporation/pan evaporation | S | 0.8–1.2 | 0.8 | 0.9 | 1.35 |
| | UM | Volume of upper layer soil moisture storage capacity (mm) | I | 5–20 | 20 | 20 | 20 |
| | LM | Volume of lower layer soil moisture storage capacity (mm) | I | 60–90 | 80 | 80 | 80 |
| | C | Conversion coefficient of deep layer evaporation | I | 0.1–0.2 | 0.15 | 0.16 | 0.16 |
| Runoff generation | WM | Volume of average soil moisture storage capacity (mm) | I | 120–200 | 160 | 223 | 163 |
| | B | The power in the curve of soil moisture storage capacity | I | 0.1–0.4 | 0.3 | 0.3 | 0.79 |
| | IM | A ratio impervious area/the area of saturated zone | I | 0.01–0.04 | 0.01 | 0.01 | 0.01 |
| Runoff partition | SM | Free water capacity in the soil surface (mm) | S | | 18 | 40 | 20 |
| | EX | The power in the curve of free water capacity in the soil surface | I | 1.0–1.5 | 0.9 | 0.9 | 1.5 |
| | KG | Outflow coefficient of free water storage to ground water | S | | 0.4 | 0.6 | 0.45 |
| | KI | Outflow coefficient of free water storage to subsurface runoff | S | | 0.35 | 0.397 | 0.4 |

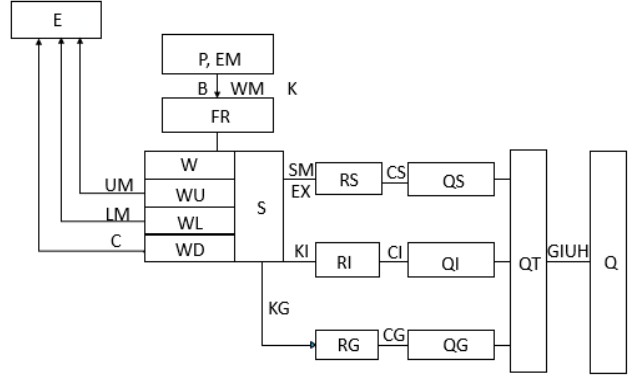

E: Evapotranspiration
P: Precipitation
EM: Evapotranspiration capability
B: Exponent of tension water capacity distribution
K: Ratio of potential evapotranspiration to the pan evapotranspiration
WM: Tension water capacity
FR: Ratio of the runoff generation area to the basin area
UM: Upper layer tension water capacity
LM: Lower layer tension water capacity
C: Deep layer evapotranspiration coefficient
W: Tension water storage
WU: Upper soil moisture
WL: Lower soil moisture
WD: Deep layer soil moisture

S: Free water storage
SM: Free water capacity
EX: Free water capacity distribution exponent
KI: Outflow ratio of free water storage to interflow
KG: Outflow ratio of free water storage to groundwater
RS: surface runoff
RI: Interflow
RG: Groundwater runoff
QS: Surface runoff inflow to river network
QI: Interflow to river network
QG: Groundwater inflow to river network
QT: Total inflow to river network
Q: Total outflow

**Figure 2.** The structure of modified Xin'anjiang (XAJ) model couple with geomorphologic instantaneous unit hydrograph (GIUH) in this study (GIUH is supposed to replace the former method to calculate the stream flow routing), and the explanation of all the items in the figure.

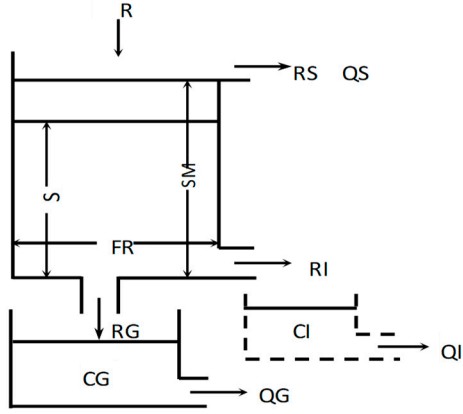

**Figure 3.** A diagrammatic sketch of the structure of "Free water reservoir" (with such a nonfigurative structure, the runoff generation can be partition based on the runoff components.).

### 3.2. GIUH based on Nash Model

#### 3.2.1. GIUH Derivation

In 1957, Nash [31] initially proposed the instantaneous unit hydrograph, which had led to a new perspective on the study of flow concentration on account of the system analysis theory. Also, in 1960, Nash [32] assumed less parameters are contained in this model and are more applicable in practical consideration. It is expressed as a mathematic formula:

$$u(t) = \left(\frac{1}{k\Gamma(n)}\right)\left(\frac{t}{k}\right)^{n-1} e^{-\frac{t}{k}} \tag{5}$$

where $u(t)$ is the ordinate of GIUH; $n$, $k$ are both of the parameters reflecting the watershed storage effects on the process from effective rainfall to discharge of watershed outlet, which separately stand for the number of conceptual free water reservoirs and a constant delineating the storage of the reservoir. $\Gamma(n)$ is the gamma function of $n$. The peak of $u(t)$, $u_p(h^{-1})$ and the time to peak, $t_p(h)$ can be derivate from the Equation (1) while using the differentiation from $u(t)$ to $t$, $\frac{du(t)}{dt} = 0$, when $t = t_p$ $u_p$ and $t_p$ are expressed, as below:

$$u_p = \frac{(n-1)^{n-1}}{K(n-1)!}e^{(1-n)} \tag{6}$$

$$t_{p1} = (n-1)k \tag{7}$$

$U$ is defined to indicate the product of $u_p$ and $t_{p1}$

$$U = u_p t_{p1} = \frac{(n-1)^n e^{(1-n)}}{(n-1)!} \tag{8}$$

It is obvious that the parameter $n$ is the only argument in the above function.

Rodriguez-Iturbe [2] came up with the expression of peak value $h_p$ and peak time $t_{p2}$ by regression analysis, when studying the characteristics of geomorphologic unit hydrograph:

$$h_p = 1.31 \times R_L^{0.43} L^{-1} v \tag{9}$$

$$t_{p2} = 0.44 R_L^{-0.38} \times (R_B/R_A)^{0.55} v^{-1} \times L_\Omega \tag{10}$$

In Equation (10), $R_A$, $R_B$, $R_L$ represent the area ratio, the bifurcation ratio, and the length ratio of a catchment, and the value of each naturally ranges from 3 to 6, 3 to 5, and 1.5 to 3.5, respectively, which are developed from Horton Laws. $L_\Omega$ (km) is a scale variable to describe the size of a watershed, which stands for the length of the highest order in the catchment. Thus, $v$ (m/s) is the mean flow velocity. The product of Equations (9) and (10), which is defined as $H$, stated as:

$$H = h_p t_{p2} = 0.58 \times (R_B/R_A)^{0.55} \tag{11}$$

Rui [33] pointed out that, when the probability density function (PDF) of GIUH turns into an exponential form, basically it is equivalent to the notional basin flow concentration model series with or parallel to linear reservoir. Consequently, the peak value and the time to peak from the nature of GIUH are supposed to be equal to which is based upon the Nash model. Therefore, when comparing the Equations (8) and (11):

$$\frac{(n-1)^n e^{(1-n)}}{(n-1)!} = 0.58 \times (R_B/R_A)^{0.55} \times R_L^{0.05} \tag{12}$$

The above formula indicates that n is merely related to the Horton three geomorphologic parameters, but irrespective of the hydrodynamic factor $v$. Alternatively, $n$ indicated the influence of watershed stream topology on flow concentration, which can be calculated from the equation, as follows:

$$n = 3.29 \times (R_B/R_A)^{0.78} \times R_L^{0.07} \tag{13}$$

As for $k$, applying the differentiation on Equation (1) to derivate the expression of $du(t)/dt = 0$, when $t = t_p$, and the Equation (7):

$$t_p = (n-1)k \tag{14}$$

$$k = \frac{0.44 R_L^{-0.38} \times (R_B/R_A)^{0.55} v^{-1} \times L}{n-1} \tag{15}$$

Here, $k$ is a scale parameter that counters both watershed topology and the average stream flow velocity $v$, showing the effect of hydraulic factor and its distribution on the flow concentration process.

### 3.2.2. Estimation of the Average Flow Velocity

The flow velocity is a crucial factor to attain the value of the indices $k$. It represents the effect of dynamic diffusion along the stream network, which is supposed to a constant throughout the basins, for the calculation of $k$. The approaches to obtain the average flow velocity have been investigated from different aspects. Two common methods are presented for comparisons. Based on the comparison of the unit hydrographs that were calculating from the two approaches, the more appropriate one is chosen to participate in the next runoff calculation.

(I) Bhaskar et al. [34] plotted the effective rainfall intensity $i_e$ and the mean flow velocity $v$ on a log-log diagram and have proposed an empirical relation between $i_e$ and $v$ by a straight line fit in three consecutive phases:

$$v = 0.72 \times i_e^{0.304}, \text{ for } i_e \leq 1.0 \text{ mm/h} \tag{16}$$

$$v = 0.98 \times i_e^{0.1841}, \text{ for } 1.0 < i_e \leq 3.0 \text{ mm/h} \tag{17}$$

$$v = 0.51 \times i_e^{0.3654}, \text{ for } i_e > 3.0 \text{ mm/h} \tag{18}$$

The corresponding velocity of each flood event can be computed by the above equation, according to the scale of the relevant mean effective precipitation intensity. Afterwards, Equation (11) is applied to calculate the parameter $k$.

(II) From the point of the definition of the average stream-flow velocity, it is explicable that $v$ is the ratio of the length of flow concentration ($L_c$) and the travel time ($t_c$). The travel time is authenticated to be equivalent to the time of flow concentration $t_c$. Hence, the mean stream flow velocity can be derived from $t_c$:

$$v = \frac{L_c}{t_c} \tag{19}$$

In 1940, Kirpich [35] deduced an empirical formula of the travel time $t_c(\text{min})$ by using the hydrologic data record in six agriculture catchments:

$$t_c = 0.0195 \times L^{0.77} \times S_m^{-0.385} \tag{20}$$

where means the average slope of the drainage area and computed by:

$S_m$: the slope of the drainage area, m/m.
$L$: the sum of the mean length of each order channel, approximate to the length of flow concentration, m, $L \approx L_c$.

Three flood events in each catchment are selected to calculate the average $i_e$ and the corresponding $v$ of each isolated flood event with the method (I). While in the method (II), it demonstrates that $v$ is independent on the rainfall, but the geomorphologic characteristic value and the value of $v$ is computed to be as a constant in every basin. The results of average $i_e$ and $v$ from two different approaches that correspond to each flood event are listed in the Table 3. The corresponding GIUH in each study region can be generated by two different ways on the calculation of the mean flow velocity. Figure 4 draws the unit hydrographs of the three study regions.

**Table 3.** Estimation of the average flow velocity in terms of two different methods (I) is calculated based on the Kirpich formula with the effective rainfall intensity in 3 flood events, and (II) is a path of computing the velocity merely dependent on the geomorphologic characteristics: the channel slope and the length of flow concentration.

| Watershed | Method (I) | | | Method (II) | | | |
|---|---|---|---|---|---|---|---|
| | Code of Flood Event | Average Effective Rainfall Intensity, $i_e$ (mm/h) | Mean Flow Velocity, $v$ (m/s) | The Mean Channel Slope of Whole Basin, $S_m$ | Length of Flow Concentration, $L_c$ (km) | Time of Concentration, $t_c$ (min) | Mean Flow Velocity, $v$ (m/s) |
| Shaowu | 19890522 | 2.94 | 1.20 | | | | |
| | 19960530 | 3.05 | 0.77 | 0.035 | 104.895 | 520.3 | 3.36 |
| | 19980302 | 2.17 | 1.13 | | | | |
| Jianyang | 19880228 | 1.59 | 1.07 | | | | |
| | 19950603 | 2.61 | 1.17 | 0.029 | 116.693 | 605.9 | 3.21 |
| | 19990715 | 2.65 | 1.17 | | | | |
| Shuiji | 19880520 | 2.43 | 1.16 | | | | |
| | 19950425 | 2.36 | 1.15 | 0.032 | 130.692 | 635.0 | 3.43 |
| | 19930615 | 1.83 | 1.10 | | | | |

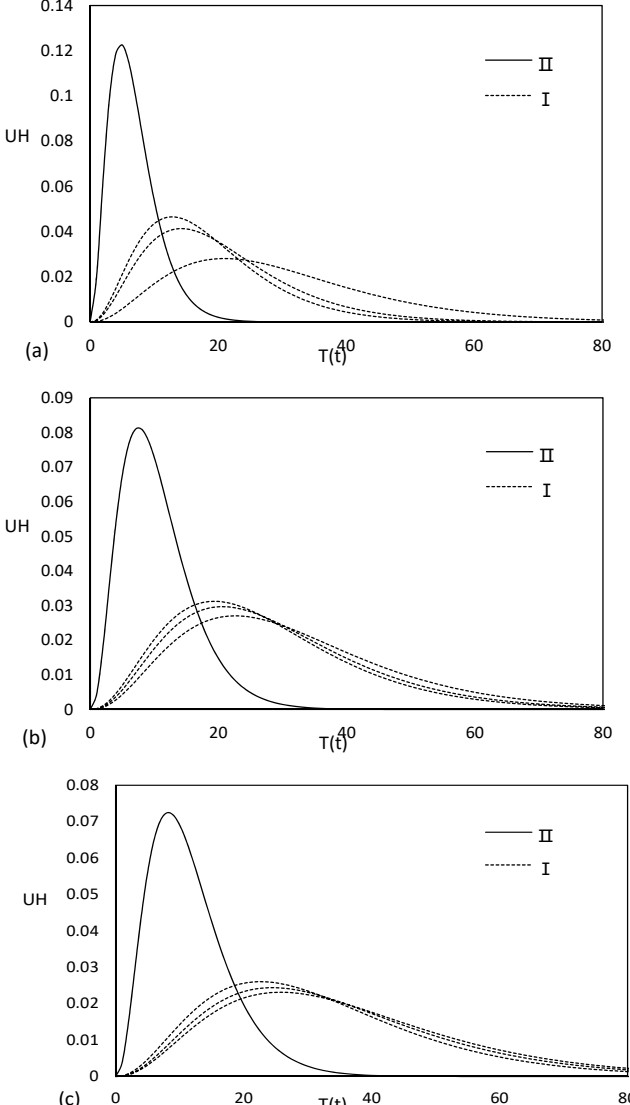

**Figure 4.** Various GIUH curves based on two different methods for Shaowu (**a**); Jianyang (**b**); and Shuiji (**c**) watersheds, with the dotted line curves standing for method (I), and the solid lines for method (II).

*3.3. Calibration and Validation on Model Parameters*

Selecting 10 historical flood events randomly, including the data of precipitation, evaporation, and flow discharge during the period of 1988 to 1999, and six floods to be as the validation calibrated the XAJ Model parameters of each catchment. Each flood event was calculated with the Xin'anjiang Model, combined with the modified GIUH model.

*3.4. Criteria on the Model Assessment*

The criteria of the model are principally applied in the simulation process to delineate the error between the observed and the simulated flow discharge at the outlet of a watershed, which are the quantitative illustration of the performance of the models. As required by the Standard Hydrological Information and Hydrological Forecasting in China [36], the model evaluation criteria are based on the statistic error function, including absolute error (AE), relative error (RE) whose upper limit is 20%, and the Nash–Sutcliffe efficiency coefficient (NSE). The time to peak error (TPE) is a criterion of AE. In respect of RE, it comprises the relative peak error (RPE) and the relative runoff depth error (RRDE).

$$\text{TPE} = t_{pc} - t_{p0} \tag{21}$$

$$\text{RPE} = \frac{(Q_C - Q_0)}{Q_0} \times 100\% \tag{22}$$

$$\text{RRDE} = \frac{(RD_C - RD_0)}{RD_0} \times 100\% \tag{23}$$

$$\text{NSE} = 1 - \frac{\sum_{i=1}^{n} (Q_{Ci} - Q_{0i})^2}{\sum_{i=1}^{n} (Q_{Ci} - \overline{Q_0})^2} \tag{24}$$

where, $Q_c$: calculated peak flow (m³/s); $Q_o$: observed peak flow (m³/s); $RD_c$: calculated runoff depth (mm); $RD_o$: observed runoff depth (mm); $Q_{oi}$ the observed discharge (m³/s); $Q_{ci}$: the calculated discharge (m³/s); $\overline{Q_0}$: is the mean of observed data $Q_{oi}$ (m³/s); and, i: time interval. It has been proved that NSE is much more superior than the deterministic coefficient as a criterion to evaluate the model performance [37].

The value of AE and RE are lower, the performance is better. While the value of NSE is larger, the flood simulation is regarded to be more fit. According to the standard of the flood estimation [36], the result of the flood simulation is qualified when value of RE is less than 20%. When the ratio of the qualified frequency to the total simulation times is greater than 0.85 and the value of NSE is larger than 0.90, the performance of the flood simulation meets the first class of forecasting standard. When the qualification ratio is between 0.70 and 0.85, along with the NSE between 0.7 and 0.9, the result satisfies the second-class standard. Otherwise, the results for flood calibration are unqualified. Table 3 summarizes the performance of all the calibrated and validated floods and the values of evaluated criteria.

In order to intuitively present the model performance, the hydrographs are drawn to compare the predicted series and the observed ones on the time series plot (Figure 5). A line fitted from the scatter plot between the measured discharge values and the simulated ones at validation stage in each basin is curved in Figure 6, with the method of least square. The relative coefficient and the slope of the fitting line are calculated to verify the model performance.

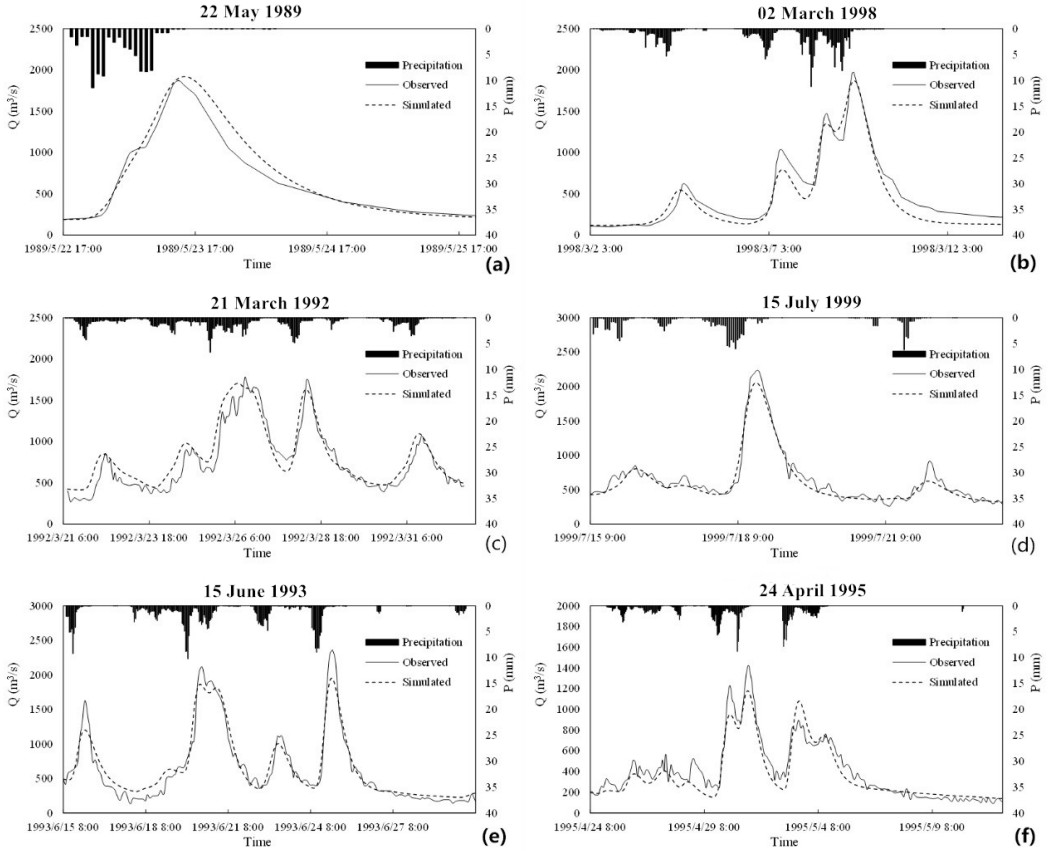

**Figure 5.** The discharge simulating graphs of two flood events in each watershed: (**a**,**b**) belong to Shaowu basin; (**c**,**d**) are for Jianyang basin; (**e**,**f**) are for Shuiji catchment; the left graphs (**a**,**c**,**e**) belong to the calibration stage and the right (**b**,**d**,**f**) are in validation step.

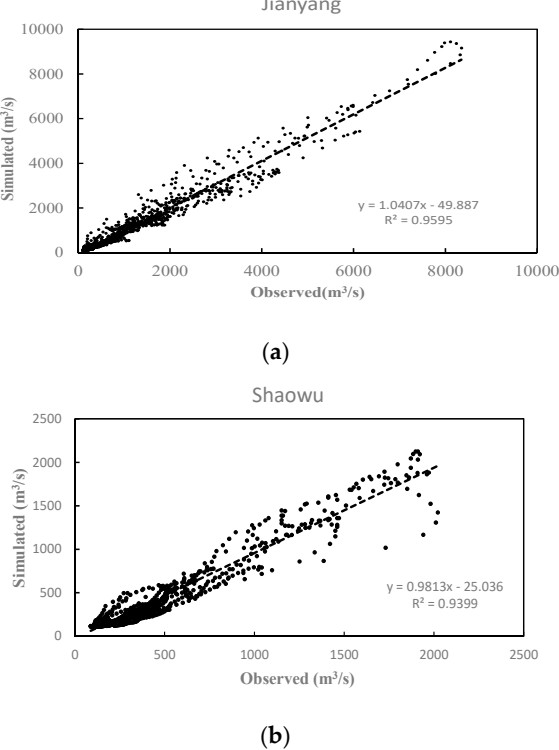

**Figure 6.** *Cont.*

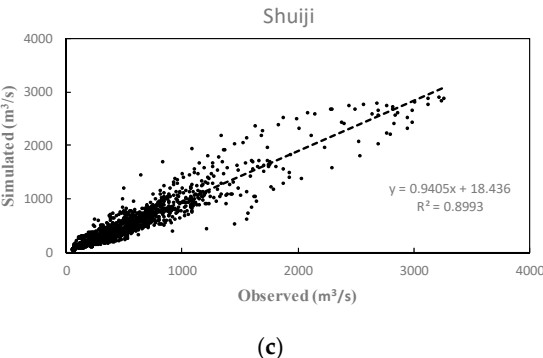

(**c**)

**Figure 6.** The fitting lines between measured and calculated runoff value in validation stage via the method of least square. (**a**–**c**) are for Jiangyang basin, Shaowu basin and Shuiji basin respectively.

## 4. Results and Discussions

In this paper, a combined rainfall-runoff model using the theory of GIUH based on Nash Model integrated with the runoff yield under the saturated zone of XAJ model has been constructed. This proposed model improves the predictive capability of the GIUH model, at the same time keeps the independence of the abundant observed data (most of the necessary parameters are extracted from the basin DEM data), increasing the applicability of the GIUH model in the ungauged or scantly gauged basins. The practicability of the modified model on flood simulation is verified. The Indices $n$ and $k$ are essential in the Nash unit hydrograph model. $n$ is supposed to be a constant, only referring to the geomorphologic parameters $R_A$, $R_L$, $R_B$, $R_L$, $R_B$, while $k$ is determined by, not only the topographic parameters, but the mean flow velocity in a watershed. There are several approaches in estimating the average flow velocity in previous research, two commonly used methods of which are chosen to generate the GIUH, meanwhile for selecting one of them to participate the final hydrologic response computation after comparison. The estimation of the average flow velocity $v$ the hydraulic factor, is operated in two methods, respectively, in virtue of the mean effective rainfall intensity $i_e$ (I), and the values of river channel characteristics (II). According to the method (I), the flow velocity can be easily obtained, which only depends on the rainfall intensity. However, it is unlikely to promptly obtain the average excess rainfall of a flood event in the real-time flood forecasting work. In contrast, method (II) is relatively an available approach that is only related to the topologic and fluvial characteristics, the flow length, and the average channel slope. Though method (I) considers different rainfall intensity rates to calculate the corresponding velocity, it neglects the impact from the important terrain and geomorphologic factors. In the meantime, the rainfall intensity variance leads to the instability of GIUH, causing the complexity of the runoff prediction based on the method (I). Hereafter, the latter one is selected to calculate the final GIUH of each basin and verify the modified GIUH model on flood simulation in this proposed study. The criteria, including the time to peak error (TPE), which is a kind of absolute error (AE), the relative peak error (RPE), and the relative runoff depth error (RRDE) in relative error (RE), and the Nash–Sutcliffe efficiency coefficient (NSE), for evaluating the performance are stated in Table 4. The summarized evaluation values of three catchments, respectively, in the calibration and validation phases are also listed in this table. The statistics that are written in bold are the outlier case showing that the flood event estimation is unsatisfied. At the calibration step, the value of NSE ranges from 0.812 to 0.941, with a mean value of 0.874 in Shaowu basin and two flood events (19920831, 19930615) are unqualified in the series, which illustrates that the result meets the second class; the average value of NSE in the Jianyang watershed reaches 0.882, 10 flood events are all qualified, satisfying the first level; as for Shuiji catchment, the NSE value ranges from 0.817 to 0.922 in this phase, with eight qualified flood events simulation. All of the XAJ parameters and the corresponding value in each catchment are listed in Table 2, and are respectively applied into the validation stage. At the validation stage, the value of criteria presents that the simulated runoff processes fit the observed well, with the average NSE value of 0.888 in three watersheds. In total, there are 11 parameters in Table 2.

It illustrates that the modified model requires much less parameters than the traditional XAJ model, correspondingly greatly reducing the uncertainty of the rainfall-runoff modelling.

**Table 4.** Summarized values of evaluation criteria and performance of 10 calibrated and 6 validated flood events in Shaowu, Jianyang, and Shuiji watershed.

| Watershed | Stage | Flood Code | TPE | RPE (%) | RRDE (%) | NSE |
|-----------|-------|-----------|-----|---------|----------|-----|
| Shaowu | Calibration | 19880228 | 1 | 1 | 20 | 0.854 |
| | | 19880512 | 0 | −19 | 6.7 | 0.892 |
| | | 19890515 | 8 | −11 | −16.3 | 0.905 |
| | | 19890522 | 1 | 3 | 8 | 0.941 |
| | | 19890621 | 0 | −16 | 3.5 | 0.924 |
| | | 19890629 | 2 | 19 | 13.3 | 0.812 |
| | | 19920321 | 0 | 17 | 12.2 | 0.851 |
| | | 19920514 | 0 | −14 | −16 | 0.877 |
| | | **19920831** | 1 | **−29** | **−22.4** | 0.834 |
| | | **19930615** | 0 | **−22** | −1.6 | 0.846 |
| | Validation | 19940501 | 0 | −5 | −9.9 | 0.832 |
| | | 19960328 | −2 | −9 | 0 | 0.856 |
| | | 19960530 | 2 | 11 | −5.9 | 0.922 |
| | | 19980215 | 0 | 1 | −7.5 | 0.851 |
| | | 19980302 | 0 | −4 | −13.9 | 0.930 |
| | | 19980509 | 6 | −9 | −10.5 | 0.898 |
| Jianyang | Calibration | 19880228 | −1 | −5 | 6.9 | 0.943 |
| | | 19880620 | 1 | −15 | 14.6 | 0.883 |
| | | 19890629 | −2 | −5 | −3.3 | 0.880 |
| | | 19920321 | −5 | −4 | 12 | 0.871 |
| | | 19920616 | −1 | −14 | 5.1 | 0.897 |
| | | 19920704 | −1 | −8 | 2.0 | 0.905 |
| | | 19920831 | 0 | −8 | 11.9 | 0.817 |
| | | 19930615 | −2 | 8 | 19.5 | 0.887 |
| | | 19930630 | −3 | −17 | −17.9 | 0.842 |
| | | 19950603 | 1 | −9 | 0 | 0.948 |
| | Validation | 19950622 | −2 | −5 | −5.8 | 0.844 |
| | | 19950626 | 0 | 6 | 0.8 | 0.914 |
| | | 19970605 | 1 | 12 | 4.7 | 0.897 |
| | | 19980608 | −2 | 13 | 2.4 | 0.944 |
| | | 19990521 | 0 | −18 | −20 | 0.876 |
| | | 19990715 | 0 | −8 | −5.2 | 0.958 |
| Shuiji | Calibration | **19880228** | 2 | −5 | **25.0** | 0.875 |
| | | 19880520 | 11 | −18 | −1.7 | 0.894 |
| | | **19880620** | 3 | **−36** | 4.2 | 0.827 |
| | | 19890520 | 1 | −17 | 5.9 | 0.882 |
| | | 19890621 | −1 | −12 | −5.9 | 0.817 |
| | | 19900629 | 1 | 7 | 11.1 | 0.864 |
| | | 19920321 | 1 | −4 | 14.6 | 0.861 |
| | | 19920514 | −1 | −11 | 10.2 | 0.912 |
| | | 19930615 | −1 | −20 | 2.1 | 0.933 |
| | | 19930502 | −2 | −7 | −2.4 | 0.890 |
| | Validation | 19940521 | 4 | −28 | −4.7 | 0.883 |
| | | 19940614 | −1 | −11 | 12 | 0.903 |
| | | 19950425 | 0 | −22 | −10.4 | 0.870 |
| | | 19950614 | −7 | −7 | −1.6 | 0.848 |
| | | 19980301 | 0 | −7 | −9.7 | 0.869 |
| | | 19990715 | 5 | −5 | −9.9 | 0.826 |

Note: Values written in bold are the outlier case showing that the flood event estimation is unsatisfied.

However, several issues should be noticed. At present, the average flow velocity can be estimated in various methods, most of which are by means of the empirical analysis, thus the velocity value can also lead to a deviation on the results of the flow simulation. Figure 5 indicates the dotted simulation hydrographs, in most cases, are lower than the bold observed curves, even after the increase of initial soil moisture. An inferred reason to explain the phenomenon is that these areas belong to the humid hilly headstream region, and because the setting of the threshold value to extract the stream channel, the complete network could not be obtained from the DEM, which causes the actual slope of the channel is higher than the calculated mean slope, meanwhile the mean flow velocity is underestimated. On the other hand, the precision of the DEM could also bring a similar influence. Six flood events are randomly selected from 48 in total to plot in Figure 5. The (a), (c), and (e) are randomly chosen from the calibrated flood events, which, respectively, represent the runoff simulation comparison curves of Shaowu, Jianyang, and Shuiji, and the (b), (d), and (f) show the hydrographs of these three catchments at the validation stage. For further analysis, the linear fit between the simulated and the observed runoff is presented in Figure 6, applying the estimation of the least squares. It clearly demonstrates that the slopes of the fitting curves in Shaowu and Shuiji are lower than 1, showing that the calculated runoff presents a state of being less than the measured value. Nevertheless, the slope of fitting line in Jianyang basin is 0.0407 greater than 1. The water conservancy project and the irrigation works, which may affect the runoff simulation, could cause it, even the calibration of the Xin'anjiang model parameters can influence the forecast results. The accuracy of the simulation is generally acceptable. However, in general, there are still some aspects of the proposed method, which need to be further improved, including the terrain and topographic data, and the calculation approach of the velocity, which is worthy of the furtherer investigation.

## 5. Conclusions

This study presents a flood forecasting approach, applying the XAJ model to calculate the flow generation, partition, and the slope flow concentration, and the optimized GIUH based on the Nash model to compute the river flow concentration. It provides a comprehensive simulation of the runoff process, keeping the advantage of GIUH being independent on the observed hydrologic data, which is suitable in ungauged or scantly gauged basins. The summarized criteria show that the modified model could achieve a satisfactory and accurate simulation result in the study of catchments. After the integration of these models, the certainty of the runoff simulation improved because the runoff generation and segmentation processes that were calculated by the XAJ model are more specific than those that are are generalized by the empirical effective rainfall calculation. Besides, topography and terrain data basically influence the GIUH, from which the information extracted is generally stable. Moreover, the application of the GIUH reduces the parameters of river flow concentration corresponding to the river channel in the traditional XAJ model, which can also lessen the simulation difficulty and uncertainty as well. In this modified model, we considered the non-uniformed vertical distribution of the soil water and the three-water-source partition to calculate the runoff yielding and partition that is exactly a vertical distributed model. It deserves a study on extending the proposed model to a spatially-distributed model by applying it in each sub-watershed, which is supposed to be more comprehensive to increase the modeling precision. In addition, the precipitation uncertainty is an essential factor in flood prediction, so in the following investigation, the rainfall error correction can be considered into the model modification.

In general, the results of the flood simulation in three watersheds clearly demonstrate a good agreement between the observed and calculated runoff. Thus, the proposed model is a potential approach for practical flood forecasting and there are still some aspects that are worthy of further research to increase the practical feasibility of the model.

**Author Contributions:** Conceptualization, Y.C. and S.Q.; Methodology, Y.C. and P.S.; Software, L.Z. and X.J.; Validation, S.M. and J.G.; Formal Analysis, Y.C. and X.J.; Investigation, L.Z. and X.J.; Resources, J.G.; Data Curation,

S.M.; Writing-Original Draft Preparation, Y.C.; Writing-Review & Editing, Y.C. and S.Q.; Visualization, P.S.; Supervision, S.Q. and P.S.; Project Administration, S.Q.; Funding Acquisition, S.Q., P.S., Y.C.

**Funding:** This research received no external funding.

**Acknowledgments:** The study is financially supported by the National Key Research and Development Program of China (grant number 2017YFC0405601), the National Natural Science Foundation of China (grant numbers 41730750, 51479062), the UK-China Critical Zone Observatory (CZO) Program (41571130071),the Graduate Student Scientific Research Innovation Projects in Jiangsu Province (grant number 2019B72014/SJKY19_0474).

**Conflicts of Interest:** The authors declare no conflict of interest.

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
