# Peer review of "Integrating XAJ Model with GIUH Based on Nash Model for Rainfall-Runoff Modelling"

_water, doi:10.3390/w11040772_

Round 1

Reviewer 1 Report

In the study “Integrating XAJ Model with GIUH Based on Nash Model for rainfall-runoff Modelling” the authors proposed a modelling framework applying the XAJ model and an optimized GIUH with Nash model to compute river flow in three basins in China. I have read the manuscript with interest. However, I have mixed feelings about it. From one side I believe that the paper is well-organized and generally of good quality. From the other side, I do not see the novelty behind the method proposed by the author. What is presented is yet another application of a lumped hydrological model combined with GIUH. As the authors mentioned in their introduction, this issue has been already addressed in the past by different researchers (e.g. see references 18, 25, 26 and 34 in the manuscript).

This raise some important issues like:

1- How does this method and study helping in advancing the scientific understanding of hydrological processes at basin scale to improve flood simulation?

2- Is this the first application of GIUH in a lumped hydrological model?

3- What is the added value of their approach if compared if already existing conceptual lumped model like HBV?

4- How the authors would extend their approach to a distributed model?

This needs to be clear since the introduction. However, the authors focused on refereeing to previous researches and application of the GIUH without proving the difference between their approach and the ones already proposed.

Below are reported some other major comments:

Major comments:

1) In the abstract, it is not clear what is the objectives and novelties of the paper. From what I read, the authors proposed a modification of the GIUH combined with the (lumped?) XAJ model to improve flood modeling. However, in the results reported in the abstract only calibration results are mentioned. What is the novelty of this study, the proposed hydrological model or the calibration procedure?

2) What is the method that the authors used to calibrate the hydrological model?

3) What do the authors mean with “But the flood simulation cannot be completed only by GIUH method”?

4) In the Introduction section, the authors provided a very comprehensive list of (quite old) studies related to GIUH estimation. However, what is missing are recent studies in which GIUH is used together with lumped (or distributed) hydrological models. In this way, it will be clear the novelty of the proposed approach if compared with others. However, if this is the first case in which GIUH and hydrological model are combined (which I seriously doubt, see references 25 and 26) the authors need to mention it.

5) A more critical analysis of the results and conclusions need to be provided. At this stage, the results and conclusions sections are quite superficial. I suggest to identify 2 or 3 research objectives of this study, include them in the introduction and then address them again in the conclusion section. This will provide a more comprehensive analysis of the results. From what I read now the objective of this study is to improve river flow estimation and conclusions are that model is good. This is valid for any paper on hydrological modelling. Authors need to be more specific on the research objectives.

6) In the conclusions, the authors stated that they proposed a flood forecasting approach. However, in this study the discussion seems to imply that the methods are to be applied for the purposes of streamflow estimation. I invite the author to change the sentence in “ proposed a method for flood estimation” and not forecasting. If forecasting was the purposes of paper it needs to clarified and justified using real forecasting input into the model. The is a difference between "prediction/forecasting", "analysis", or "reanalysis" and authors must be aware of that.

7) Include limitations of this study and recommendations in the Conclusion section.

8) There are a handful of places where more specific language would go a long way towards clarity.

9) Figures are difficult to read, improve the quality

10) Tables and figures captions are inadequate in many places. Generally, I like to see all the terms used in the figure defined in the caption so that the paper can be refreshed in a readers memory just by reading the figures.

Other comments:

1) Line 38-39: “especially in some areas with insufficient hydrologic data” How is the lack of data affecting the flood damages? Include more details

2) Line 328, what the authors mean with “practicably”??

3) Line 351, “and at the validation...”, Do not start a sentence with “and”.

4) Line 372, replace “It could be caused” with “This could be...”

5) Line 393-394, “pretty accurate results” please be more specific. Also, pretty accurate compare to what? Did the authors compared their method with other widely used conceptual lumped model (e.g. HBV)?

6) Line 396, “easy to acquire”, what do you mean?

7) Lines 399-402. How the author can state that their approach can be potentially used in flood forecasting if they use it for flood simulations? Did they include uncertainty in precipitation input? This is essential for a correct forecast of river flood.

Author Response

Dear Reviewer:

We are truly grateful to your valuable comments and thoughtful suggestions. Based on these comments and suggestions, we have made careful revisions on the original manuscript. In the attachment, we list our point by point response ( written in red) to your comments, and all the revision are labeled in the revised manuscript. 

Kind regards

Yingbing Chen

Reviewer 2 Report

All comments are included in the attached word file.

Author Response

Dear reviewer:
Many thanks for your insightful comments and suggestions of the referees. We have made the corresponding response point by point according to your comments. Words in red are the changes I have made in the attachment.

Yours sincerely,

Yingbing Chen

Round 2

Reviewer 1 Report

I thank the authors for including all my comments and suggestions in the updated version of the manuscript. I do not have any further comment

Author Response

Dear reviewer,

Thank you so much for your effort for improving our paper quality and your final approval of our work. 

Best regards,

Yingbing Chen

Reviewer 2 Report

Overall, you have corrected numerous observations previously made, but there are still a few remarks to be made:

Figure 1 - it has undergone improvements, but is not final. The legend was not edited, which it should be, and also, I strongly recommend adding scale bars to all 3 maps, for improved cartographic consistency.

The 11 year time span is a problematic matter. Considering you do not have access to official, national data, have you tried finding alternative sources?

In the original manuscript version, you have acknowledged that “But in general, there are still some aspects of the proposed method which need to be further improved, including the terrain and topographic data, and the calculation approach of the velocity, which is worthy of the furtherer investigation.”. I have also stated that “Analysis on velocity calculations should be improved, while integrating topographical and corrected/validated terrain data, which accounts for technology with which it was acquired”. I do not consider that the clarifications are up to the observations, I have made previously. This is due to the fact that velocity is a critical factor of flood analysis. For example, in Table 3, there are threefold differences between mean flow velocities associated to the 2 different applied methods. This should be further detailed in chapter 3.2.2 or 4, as you only describe the workflow, and do not discuss the compared results sufficiently, generated by the 2 methods, and why they are so different, and how they influence the model. I recommend the discussion used in chapter 4 to be extended for a better understanding of all the issues regarding flow velocity in the present study, especially considering that you stated the use of empirical methods, in this regard (“At present, the average flow velocity can be estimated in various methods, most of which are by means of the empirical analysis, thus the velocity value can also lead to a deviation on the results of the flow simulation.”), and there is no clear comparison discussion for the highly different flow velocity values.

Author Response

Dear reviewer,

Thank you so much for your patience and your effort for helping us to improve the paper quality, the attachment which contains the point-by-point replies to your comments is uploaded to the system, so is the revised manuscript. Hope it can get your professional approval.

Best regards,

Yingbing Chen

Round 3

Reviewer 2 Report

Dear authors,

Thank you for your explanations, which are highly appreciate and considerate. I understand the restrictions related to the data acquisition process, and I am looking forward for your future studies, which you have mentioned in your comments. Please consider improving future studies, regarding runoff, and included models, in order to publish manuscript in a form that emphasizes a finished analysis, in favor of studies which are not completed.

Best regards